# AQP3 Facilitates Proliferation and Adipogenic Differentiation of Porcine Intramuscular Adipocytes

**DOI:** 10.3390/genes11040453

**Published:** 2020-04-22

**Authors:** Xiaoyu Wang, Jing Yang, Ying Yao, Xin’E Shi, Gongshe Yang, Xiao Li

**Affiliations:** Key Laboratory of Animal Genetics, Breeding and Reproduction of Shaanxi Province, College of Animal Science and Technology, Northwest A & F University, Yangling 712100, China; wangxy067@163.com (X.W.); yangjing@nwsuaf.edu.cn (J.Y.); yaoying@nwsuaf.edu.cn (Y.Y.); xineshi@163.com (X.S.); gsyang999@hotmail.com (G.Y.)

**Keywords:** AQP3, pig, intramuscular fat, adipogenesis, proliferation

## Abstract

The meat quality of animal products is closely related to the intramuscular fat content. Aquaglyceroporin (AQP) defines a class of water/glycerol channels that primarily facilitate the passive transport of glycerol and water across biological membranes. In this study, the AQP3 protein of the AQP family was mainly studied in the adipogenic function of intramuscular adipocytes in pigs. Here, we found that AQP3 was increased at both mRNA and protein levels upon adipogenic stimuli in porcine intramuscular adipocytes in vitro. Western blot results showed knockdown of AQP3 by siRNA significantly suppressed the expression of adipogenic genes (PPARγ, aP2, etc.), repressed Akt phosphorylation, as well as reducing lipid accumulation. Furthermore, deletion of AQP3 by siRNA significantly downregulated expression of cell cycle genes (cyclin D, E), and decreased the number of EdU-positive cells as well as cell viability. Collectively, our data indicate that AQP3 is of great importance in both adipogenic differentiation and proliferation in intramuscular adipocytes, providing a potential target for modulating fat infiltration in skeletal muscles.

## 1. Introduction

Ectopic fat deposition in skeletal muscle has attracted increasing attention in recent decades. In humans, excessive fat accumulation in skeletal muscle always represents muscle weakness, myopathy, and metabolic diseases such as obesity, diabetes, coronary heart disease, etc. [1]. Conversely, in livestock animals (cattle, pig, sheep, etc.), distribution of intramuscular fat (IMF), referred to as marbling by customers, is closely related to meat quality, and a moderate increase in IMF benefits the taste and flavor of meat products [2]. The high content of intramuscular fat can increase the tenderness and flavor of pork [3]. Marbling develops either by an increase in adipocyte number, or adipocyte volume, or both. Thus, the molecular mechanisms underlying the proliferation and differentiation of intramuscular adipocytes deserve further study.

Aquaglyceroporin (AQP) refers to a subgroup of aquaporins that conduct glycerol, water, and other small polar solutes in response to osmotic gradients. Glycerol is a necessary constituent of triglyceride (TG) backbones, and glycerol uptake together with release across the plasma membrane are two key steps in triglyceride synthesis (lipogenesis) and hydrolysis (lipolysis) in adipose, liver, and other metabolic organs, thus AQPs have emerged as key players in adipose biology and the development of obesity [4]. In this context, AQP7, the first identified AQP in adipose tissue, has been emerging as an important player in whole-body metabolism and the progress of obesity and diabetes [5]. In recent studies, expression of AQP3, AQP9, AQP10, and AQP11 has been detected in cultured adipocytes and adipose tissues [6,7,8]. Studies have shown that AQP3 can promote the transport of cellular glycerol and has affinity for glycerol [9]. AQP3 has been reported to regulate PPARα by adiponectin in hepatic stellate cells (HSC) [10]. The classical secretion factor, leptin, of adipocytes can improve systemic obesity and fatty liver in mice via AQP3 in ob/ob mice [11]. AQP3 mRNA expression in human adipose tissue was reported in another study [12]. These signs indicate that AQP3 may also have a regulatory role in IMF.

Our RNA-seq screening revealed that AQP3 [13], in addition to the well-known AQP7, was upregulated during adipogenic differentiation in porcine intramuscular preadipocytes (PIPAs), indicating a potential role of AQP3 in porcine IMF deposition. Work on the MDA-MB-231 breast cancer cell line has shown that knockdown of AQP3 modestly decreases water permeability (17%), but markedly decreases glycerol permeability (77%) [14], indicating AQP3 might be more permeable to glycerol. Currently, the most well-characterized role of AQP3 is the promotion of cancer metastasis, for AQP3 is abnormally escalated in various kinds of cancers [15] and knockdown of the AQP3 gene could significantly decrease cell proliferation, and increase cell death or apoptosis in cancer cells [14,16,17]. In addition, AQP3 deficiency can cause proliferation disorders and metabolic inhibition in gastric cancer cells [18]. More interestingly, another study showed that AQP3 in gastric cancer cells caused apoptosis in gastric cancer cells by downregulating cellular glycerol intake and inhibiting downstream adipogenesis [19]. These data encouraged us to explore the effects of AQP3 on adipogenic differentiation, lipid deposition, and proliferation, using porcine intramuscular preadipocytes as a model.

## 2. Materials and Methods

### 2.1. Animal Care

Piglets in our study were obtained from the experimental plot of Northwest A&F University (Yangling, China). Pigs were reared under standard light and temperature conditions and allowed food and water ad libitum. This project was approved by the Institutional Animal Care and Use Committee of Northwest A&F University.

### 2.2. Cell Culture

Porcine intramuscular preadipocytes were isolated from the longissimus dorsi muscle of 3-day-old piglets as previously described [20]. The specific method was as follows: LD (longissimus dorsi) muscles were quickly excised, rinsed twice in sterile pre-cooled phosphate-buffered saline (PBS), and then cut into 1 mm^3^ sections. Muscle fragments were incubated in Dulbecco’s Modified Eagle’s Medium/F12 (DMEM/F12; Hyclone, Logan, UT, USA) containing 0.1% I type collagenase (270 U/mg; Gibco, Carlsbad, CA) for 1.5 h in a 37 °C water bath, with continuous shaking. The products were then sequentially passed through a 70 mesh (212 μm) and then a 200 mesh (75 μm) to obtain single cells. The cells were seeded in a dish with DMEM/F12 medium containing 10% fetal bovine serum (Gibco, Grand Island, NY, USA). After 2 h, we changed the medium to keep only adherent cells. In the proliferating stage, the primary preadipocytes were cultured in DMEM/F12 (Gibco, Grand Island, NY, USA) containing 10% FBS (Invitrogen, Carlsbad, CA, USA).

To induce adipogenic differentiation, when cells achieved 100% confluence, a mixture containing 10% FBS, 5 μg/mL (872 nM) insulin, 1 μM dexamethasone (DEX), and 0.5 mM isobutyl methylxanthine (IBMX; Sigma-Aldrich, St Louis, MO) was used to induce adipogenic differentiation. Two days later, a DMEM/F12 medium containing 10% fetal bovine serum (FBS) and 5 μg/mL (872 nM) insulin was changed to maintain differentiation.

### 2.3. Transfection with siRNA

Oligonucleotides of AQP3 siRNAs (forward: CCCUUAUCCUCGUGAUGUUTT, reverse: AACAUCACGAGGAUAAGGGTT) and NC (negative control, forward: UUCUCCGAACGUGUCACGUTT, reverse: ACGUGACACGUUCGGAGAATT) were obtained from GenePharma (Shanghai, China). Transfection was performed with Lipofectamine^®^ RNAiMAX Reagent (ThermoFisher, Waltham, MA, USA) when cells reached proper confluence (40–50% for proliferation test, 70–80% for differentiation test). Negative control and siRNA were added to Opti-MEM (Gibco) and mixed with Lipofectamine^®^ RNAiMAX Reagent. The mixture was allowed to stand for 20 min before being added to the cell culture plate. The culture medium was replaced after 24 h. The final concentration of siRNA or negative control was 50 nM. Cells were changed into fresh growth medium 24 h post-transfection. 

### 2.4. RNA Isolation and RT-qPCR

Total RNA was purified using Trizol (TaKaRa Bio, Inc., Dalian, China) and was subjected to reverse transcription using the PrimeScriptTM RT reagent Kit (TaKaRa Bio, Inc., Dalian, China). A tissue sample (0.5 g) from a 180-day-old pig was weighed, and high-throughput grinding of the tissue sample in Trizol infiltration extracted total RNA. Each experimental group was subjected to the reverse transcription reaction with 500 ng of RNA. cDNA was subjected to the Multicolor Real-Time PCR detection system (iQ5, Bio-Rad Laboratories, Inc., Hercules, CA, USA) with SYBR Premix Ex TaqTM II kit (TaKaRa Bio, Inc., Dalian, China). The procedure of PCR reaction was pre-denaturation for 5 min, followed by denaturation for 10 s, annealing for 30 s, and extension of 30 s for 35 cycles. Primers targeting AQP3 [21], PPARγ [22] (peroxisome proliferator activated receptor γ), FABP4 [23] (adipocyte fatty-acid binding protein 4), mGPAT [24] (mitochondrial glycerol-3-phosphate acyltransferase), Perilipin 1 [25], cyclin B [26], and β-actin [27] were picked out in previous reports. Primers for SCD (stearoyl-CoA desaturase), CD36, C/EBPα (CCAAT/enhancer binding protein α), ELOVL6 (elongase of long-chain fatty acids family), FASN (fatty acid synthetase), ACACA (acetyl-CoA carboxylase), DGAT2 (diacylglycerol O-acyltransferase 2), cyclin E, and cyclin D were designed online (https://www.ncbi.nlm.nih.gov/tools/primer-blast/) and synthesized by Sangon Biotech (Shanghai, China). Relative expression of each gene was calculated using the 2−∆∆ Ct method, using β-actin as the internal control. Sequences for all primers are shown in Table 1.

### 2.5. Western Blot

Cells were scraped with RIPA (Radio Immunoprecipitation Assay) buffer (Beyotime, Shanghai, China) and lysates were subjected to SDS-PAGE and transferred to the PVDF (Polyvinylidene fluoride, Millipore, Burlington, MA, USA). Polyacrylamide gels were used to separate and mark proteins of different sizes. The proteins were then transferred to a PVDF membrane. Next, the membrane was soaked in 5% skim milk for 2 h and then incubated with primary antibodies overnight at 4 °C. After that, membranes were washed in Tris-buffered saline with Tween 20 and subsequently incubated with horseradish peroxidase-conjugated secondary antibodies. Finally, the stripes of target proteins were visualized by the enhanced chemiluminescent substrate (Millipore, MA) and observed using Gel Doc XR System (Bio-Rad). The densities of the brands were analyzed using Image Lab software (Bio-Rad). Target proteins were probed with primary antibodies (anti-AQP3, ab125219, Abcam, 1:1000; PPARγ, #2435, CST, 1:1000; FASN, sc-20140, Santa Cruz, 1:200; FABP4, sc-18661, Santa Cruz, 1:200; Akt, sc-8312, Santa Cruz, 1:200; p-Akt, sc-7985-R, Santa Cruz, 1:200; cyclin D, sc-753, Santa Cruz, 1:500; cyclin E, sc-247, Santa Cruz, 1:500; β-actin, sc-130656 Santa Cruz, 1:1000).

### 2.6. Oil Red O Staining

The well-differentiated cells were washed twice with PBS and fixed with 4% paraformaldehyde for 30 min, and then incubated with 1% filtered Oil Red O solution for 5 min. The stained lipid droplets in the cells were photographed (Nikon TE2000 microscope, Tokyo, Japan). For quantitative analysis, cellular Oil Red O was extracted by isopropanol and optical absorbance was detected at 510 nm.

### 2.7. EdU Staining

EdU assay was conducted with a Cell-Light^TM^ EdU (5-ethynyl-2′-deoxyuridine) Apollo^®^567 In Vitro Imaging Kit (RiboBio, Guangzhou, China) as per the manufacturer’s instructions. Porcine intramuscular preadipocytes in growth medium were incubated with 50 mM EdU for 2 h, and then fixed by paraformaldehyde. Then cells were labeled with Apollo reaction solution, and the nuclei were stained with Hoechst 33,342 (Thermo Fisher Scientific, Waltham, MA, USA). Cells were visualized using a Nikon TE2000 microscope (Nikon, Tokyo, Japan), and the images were processed with Image J software by the National Institutes of Health (NIH).

### 2.8. CCK-8 Assay

Porcine intramuscular preadipocytes were seeded on a 96-well plate at a density of 1 × 10^3^ per well. Some 24 h later, 10% CCK-8 solution (Vazyme, Nanjing, China) was added, and after 4 h incubation, the absorbance was measured at 490 nm.

### 2.9. Statistical Analysis

All experiments were carried out in triplicate and the results were analyzed by one-way analysis of variance (ANOVA) using SPSS 18 software (SPSS Inc., Chicago, IL, USA). *P* < 0.05 was set as statistical significance. Data were presented as mean ± standard error (SE).

## 3. Results

### 3.1. AQP3 Is Upregulated during Adipogenesis

In order to explore the expression pattern of AQP3 gene in pig adipose tissue and PIPAs we selected 180-day-old pig tissue to test the expression of AQP3. RT-qPCR results showed that AQP3 is highly expressed in adipose tissue (Figure 1A). In subcutaneous adipose tissue of pigs of different ages, the expression of AQP3 reached the highest at 30 days of age, and then began to decline (Figure 1B).

In in vitro cell culture experiments, transcripts of AQP3 in PIPAs were rapidly increased upon adipogenic stimuli, reached a peak at 4 d post-differentiation, and then gradually decreased (Figure 2A). The expression of AQP3 proteins showed the same pattern (Figure 2B). For reference, the expression of PPARγ (Figure 2C), FASN (Figure 2D), and aP2 (Figure 2E) throughout the adipogenic process was profiled to represent the efficient differentiation of PIPAs in vitro. The data indicated a promising role of AQP3 in adipogenesis.

### 3.2. Knockdown of AQP3 Blunts Adipogenesis

In view of the rising trend of AQP3 in adipocyte differentiation, siRNAs were employed to explore the role of AQP3 on adipogenic differentiation. Three siRNAs were designed, and only siRNA-1 showed >70% knockdown efficiency 24 h post-transfection (Figure 3A), and still significantly reduced AQP3 expression 48 h, 4 d, and 8 d post-differentiation (Figure 3B). Thus, siRNA-1 was used in the following study. RT-qPCR results showed that AQP3 siRNA significantly inhibited the expression of adipogenic markers, such as PPARγ, aP2, ACACA, SCD, DGAT2, mGPAT, ELOVL6, and FASN 4d post-differentiation (Figure 3C), and the genes (except PPARγ) detected above were still significantly downregulated 8 d post-differentiation (Figure 3D). Western blot results presented the expression of PPARγ, aP2, and FASN at protein levels and phosphorylated Akt (Figure 3E), and the gray level analysis showed that these proteins were significantly decreased (Figure 3F). Oil Red O staining showed that AQP3 siRNA significantly repressed triglyceride accumulation in intramuscular adipocytes (Figure 3G,H). These data indicated that AQP3 was essential for adipogenesis and lipid accumulation in PIPAs.

### 3.3. AQP3 Deletion Inhibits Proliferation

AQP3 siRNA could significantly downregulate AQP3 mRNA expression in the proliferating porcine intramuscular preadipocytes 24 h post-transfection (Figure 4A), indicating that this siRNA can be used to inhibit the expression of AQP3 during the proliferation phase. The results of RT-qPCR showed that siRNA repressed the expression of cyclin B, cyclin D, cyclin E, and CDK4 mRNAs 48 h post-transfection (Figure 4B). Consistently, cyclin B, cyclin D, and proliferating cell nuclear antigen (PCNA) were significantly repressed by AQP3 siRNA at the protein level too (Figure 4C,D). Additionally, transfection of AQP3 siRNA significantly reduced the ratio of EdU-positive cells (Figure 4E,F) and cell viability (Figure 4G). In summary, the above results show that during the proliferation phase, AQP3 exerts this effect to promote cell proliferation.

## 4. Discussion

The relationship between AQP3 and adipose tissues or cells has been ignored for a long time, and its expression in adipose tissues or adipocytes remains ambiguous. In pioneering work, AQP3 mRNA was undetectable in adipose tissues of Meishan pigs by semi-quantitative RT-PCR [28]. Meanwhile, there were other studies that did not support the presence of AQP3 in mouse [29,30] or human adipose tissue [31]. However, mRNAs and proteins of AQP3 were later detected in human stroma vascular fraction of omental, subcutaneous adipocyte tissue, and also in freshly isolated adipocytes [6]. Additionally, the expression of AQP3 was confirmed in murine 3T3-L1 cell line [32,33]. The inconsistent reports might be due to the relatively lower expression levels of AQP3 in mature adipose tissues [6,12].

In the present study, RT-qPCR and Western blot uncovered the upregulation of AQP3 during the process of adipogenic differentiation, indicating that AQP3 may also play a regulatory role in intramuscular adipocytes. Furthermore, AQP3 knockdown by siRNA leads to reduced expression of adipogenic and lipogenic genes and defects of TG storage in porcine intramuscular preadipocytes. After AQP3 was silenced, the expression of the classic adipogenic factor PPARγ was significantly inhibited. PPARγ is an indispensable transcription factor for adipocyte differentiation [34], and a previous study has shown that AQP3 is the target of PPARγ in murine adipose cell line [35] and hepatic stellate cells [36]. The level of phosphorylation of AKT was also decreased, which indicates that the downstream signal of the classic insulin signaling pathway is weakened. At the same time, the expression levels of other adipogenic marker genes aP2, ACACA, etc. were also decreased, indicating that the silencing of AQP3 inhibits the adipogenic differentiation of porcine intramuscular preadipocytes from the overall gene expression level. A previous study has also shown that AQP3 is more permeable to glycerol compared with water [14], and AQP3 was increased in LPS-induced adipogenesis [32] and triglyceride sedimentation [37]. A combination of our and others’ work supports a pivotal role of AQP3 in lipid accumulation in adipocytes. 

Besides, AQP3 also has the potential to regulate cell proliferation. Accumulated studies have shown that AQPs are involved in tumor metastasis [15]. AQP3 and AQP5 can be used as new markers for breast cancer [38]. Another study showed that overexpression of AQP3 in mammalian cells can promote cell proliferation efficiency and cell cycle transition [39]. In our study, AQP3 siRNA weakened the proliferation ability of porcine intramuscular preadipocytes, which was reflected in the downregulation of genes such as cyclin B and cyclin D, and a decrease in the number of EDU^+^ cells. This shows that AQP3 can not only regulate cell proliferation in cancer cells, but also in intramuscular preadipocytes. For IMF, the proliferation and differentiation of adipocytes are two very important fat deposition processes [40]. AQP3 shows regulatory capacity in both proliferation and differentiation, which further illustrates that AQP3 is a key gene for adipocytes.

In the AQPs family, AQP7 can promote the loss of glycerol from adipocytes and inhibit the accumulation of fat, while AQP9 is responsible for the uptake of glycerin [41]. Additionally, the weakening of AQP5 can inhibit the late adipogenesis of 3T3-L1 cells [42]. Compared with other AQP genes, the present study proves that AQP3 can regulate fat deposition in two ways, namely adipocyte proliferation and differentiation, revealing that AQP3 is likely to be a new key factor in regulating fat deposition, which supplements the regulation of adipocytes by AQPs. 

## 5. Conclusions

Collectively, our work has identified AQP3 as a novel and essential modulator in proliferation, differentiation, and lipid accumulation in intramuscular adipocytes, providing a new theoretical basis for the regulation of skeletal muscle ectopic fat deposition. This conclusion may be due to the involvement of AQP3 in the transport of glycerol, but further research is needed.

## Figures and Tables

**Figure 1 genes-11-00453-f001:**
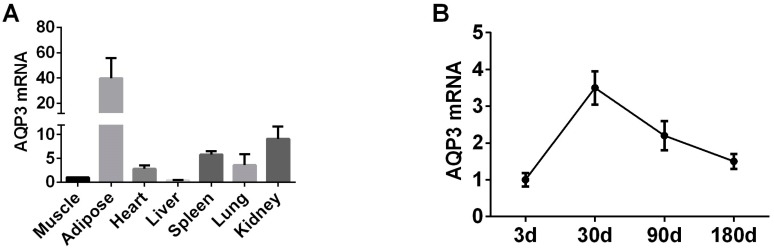
AQP3 expression pattern in pigs in vivo by RT-qPCR. (**A**) AQP3 pig tissue expression profile. (**B**) AQP3 expression during pig growth by RT-qPCR. β-actin was used as internal reference gene. *n* = 3.

**Figure 2 genes-11-00453-f002:**
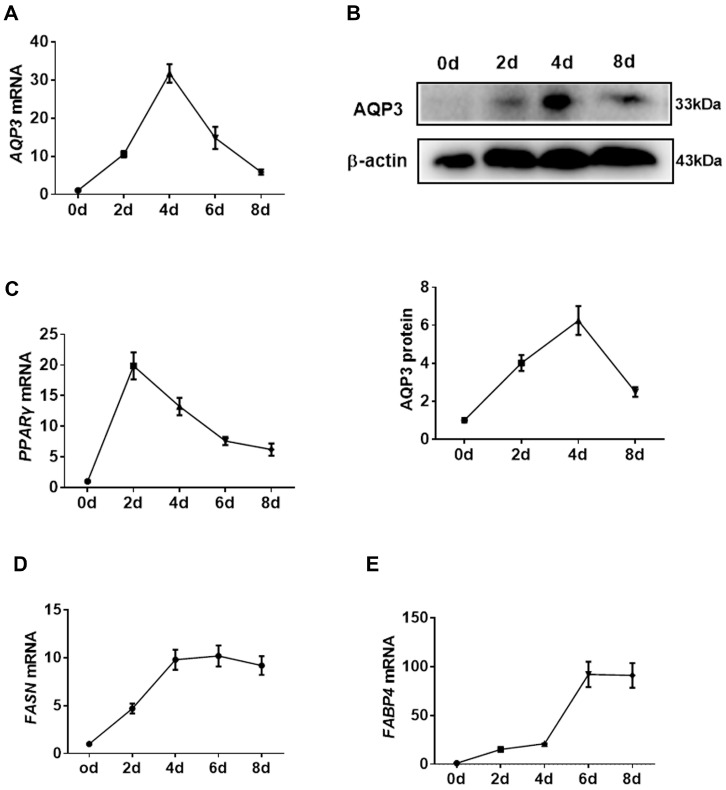
Expression pattern of porcine AQP3 in adipogenesis. The expression of AQP3 during adipogenesis was detected by RT-qPCR (**A**) and Western blot (**B**). The expression of PPARγ (**C**), FASN (**D**), and aP2 (**E**) was detected by RT-qPCR. β-actin was used as internal reference gene. *n* = 3.

**Figure 3 genes-11-00453-f003:**
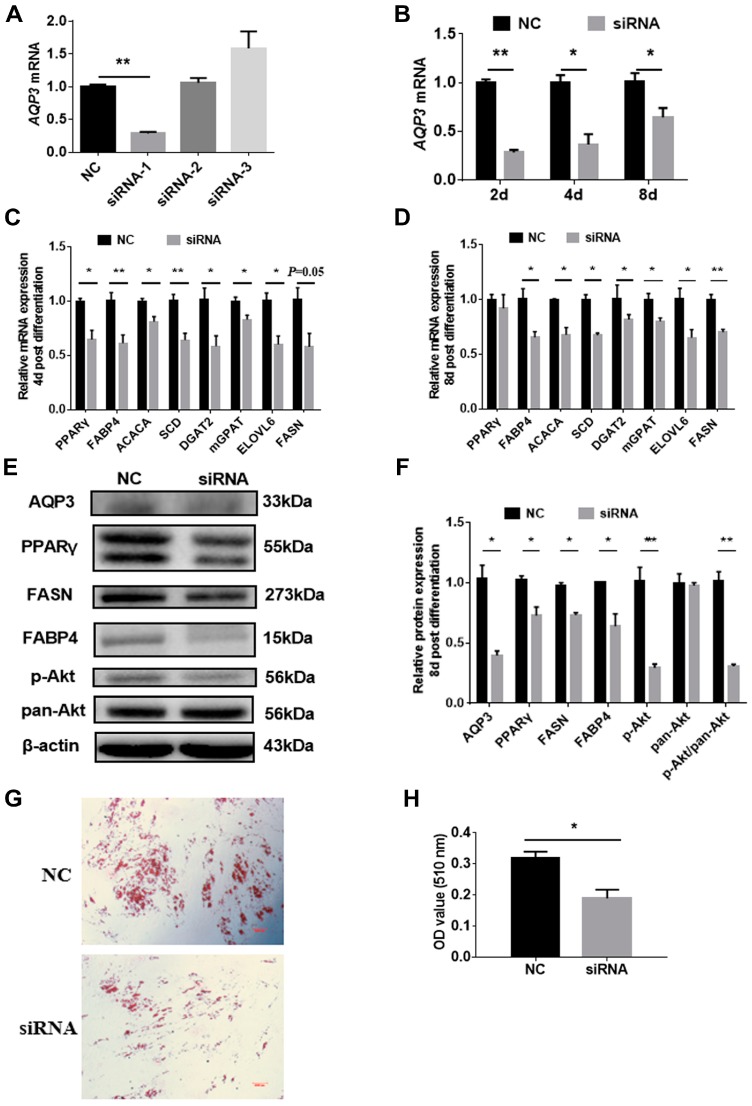
AQP3 silence repressed adipogenic differentiation in porcine intramuscular adipocytes. Cells were transfected with 3 candidate siRNAs targeting AQP3 when reaching 70–80% confluence, and only siRNA-1 could decrease AQP3 transcripts by 70% (**A**) 24 h post-transfection, and it also significantly repressed AQP3 expression 48 h, 4 d, and 8 d post-differentiation (**B**). Expression of adipogenic and lipogenic genes 4 d and 8 d post-differentiation was detected by RT-qPCR, using β-actin as reference gene (**C**). Expression of adipogenesis-related genes 8 d post-differentiation was detected by Western blot (**D**). Western blot images (**E**) and gray analysis statistics (**F**) of PPARγ, aP2, FASN, and Akt in PIPAs. Lipid accumulation was tested by Oil Red O staining (**G**) and quantified by isopropanol extraction (**H**). *n* = 3; * *p* < 0.05, ** *p* < 0.01.

**Figure 4 genes-11-00453-f004:**
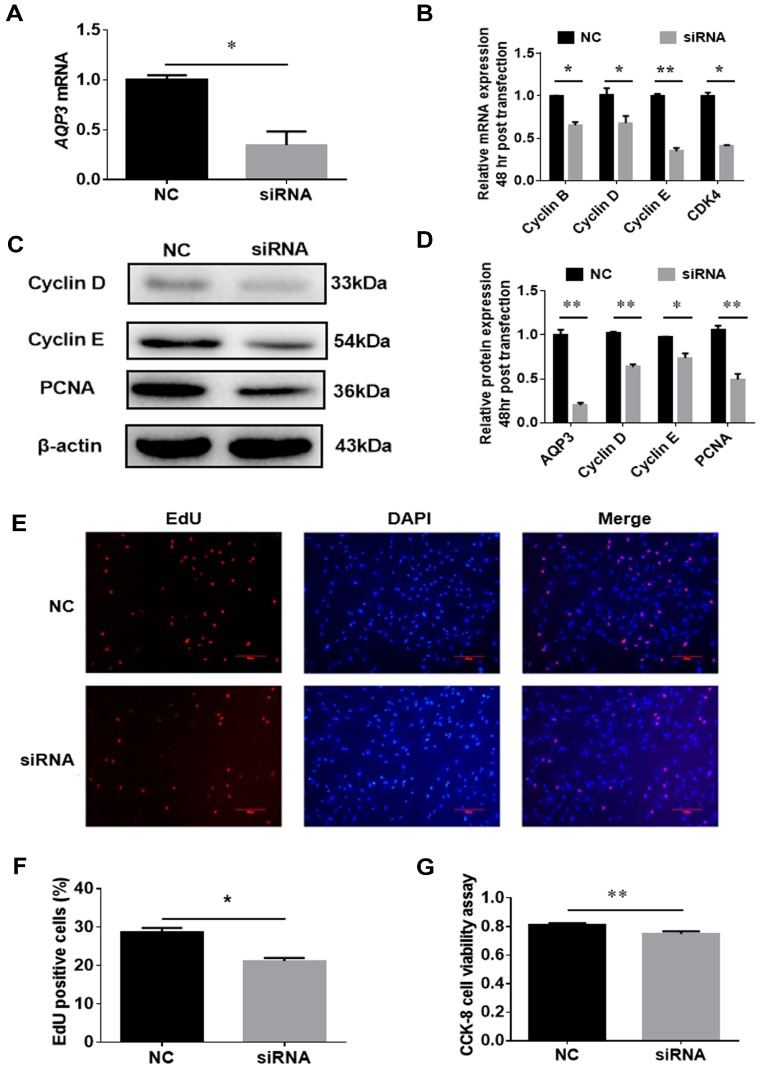
Knockdown of AQP3 inhibited the proliferation of porcine intramuscular adipocytes. Cells were transfected with AQP3 siRNA at 40–50% confluence, and the interference effect was over 70% 24 h post-transfection (**A**). Cell cycle genes were analyzed by RT-qPCR (**B**) and Western blot (**C**,**D**). EdU staining was captured (**E**) and EdU-positive cells were counted to monitor the proliferation of cells (**F**). CCK-8 was adopted to check cell viability (**G**). *n* = 3. CDK4, cyclin-dependent kinase 4; PCNA, proliferating cell nuclear antigen; DAPI, 4′,6-diamidino-2-phenylindole; * *p* < 0.05, ** *p* < 0.01.

**Table 1 genes-11-00453-t001:** Primer sequences for real-time qPCR.

Gene	Accession Number	Primer Sequences	Production Length/bp
**AQP3**	NM_001110172.1	F: CACCTCCATGGGCTTCAACT	278
		R: TGCCCATTCGCATCTACTCC	
**PPARγ**	NM_214379	F: AGGACTACCAAAGTGCCATCAAA	142
		R: GAGGCTTTATCCCCACAGACAC	
**aP2**	NM_001002817.1	F: GAGCACCATAACCTTAGATGGA	121
		R: AAATTCTGGTAGCCGTGACA	
**FASN**	NM_001099930.1	F: GTCCTGCTGAAGCCTAACTC	206
		R: TCCTTGGAACCGTCTGTG	
**SCD**	NM_213781.1	F: ACAAGAGGCCAAGACAAGTTCC	142
		R: GCTGTAGGGAATGCTGGTTAGTTT	
**ACACA**	NM_001114269.1	F: TCCCAGTGCAAGCAGTATG	211
		R: TGCCAATCCACACGAAGAC	
**mGPAT**	XM_005671462.3	F: ACTATCTCCTGCTCACTTTCA	146
		R: CGTCTCATCTAGCCTCCGTC	
**CD36**	NM_001044622.1	F: ATCGTGCCTATCCTCTGG	103
		R: CCAGGCCAAGGAGGTTAA	
**C/EBPα**	XM_003127015.4	F: AACAACTGAGCCGCGAACTG	181
		R: GCTCCGGCAGTCTTGAGAT	
**DGAT2**	NM_001160080.1	F: GCAGGTGATCTTTGAGGAGG	140
		R: GCTTGGAGTAGGGCATGAG	
**ELOVL6**	XM_021100708.1	F: ACCACATCACTGTGCTCCTC	95
		R: CGAGTGCACGCCATAGTTCA	
**Cyclin B**	NM_001170768.1	F: AATCCCTTCTTGTGGTTA	104
		R: CTTAGATGTGGCATACTTG	
**Cyclin E**	XM_005653265.2	F: AGAAGGAAAGGGATGCGAAGG	173
		R: CCAAGGCTGATTGCCACACT	
**Cyclin D**	XM_021082686.1	F: TACACCGACAACTCCATCCG	224
		R: GAGGGCGGGTTGGAAATGAA	
**β-actin**	XM_021086047.1	F: GGACTTCGAGCAGGAGATGG	138
		R: AGGAAGGAGGGCTGGAAGAG	

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
