# Peer review of "AQP3 Facilitates Proliferation and Adipogenic Differentiation of Porcine Intramuscular Adipocytes"

_genes, 2020, doi:10.3390/genes11040453_

Round 1
Reviewer 1 Report
The presented article "AQP3 facilitates proliferation and adipogenic differentiation of porcine intramuscular adipocytes" focus its attention on AQP3 role in signalling environment regarding intramuscular adipocytes in pigs. The presented topic is engaging and sheds new light on this subject. Authors used state-of-the-art techniques and kept their research plan in a precise and coherent design.
I would consider this article allowed for publication, after revision of remarks by the authors, which are listed below.
line 18: should be either "et alia" or "et al."
line 30: excessive space between "fat" and "(IMF)"
line 42-43: the sentence "(...) detected in cultured adipocytes and adipose tissue." is followed by reference [6], that is a review publication instead of the research article. Stronger emphasis on that fact could be beneficial for the clarity of the text. Additionally, authors could distinguish between research focused on pre-adipocytes and adipocytes, that are more difficult to culture.
line 60: Should be "mm3" instead of "mm3".
line 63 and 64: Missing "µm".
Line 69: All necessary abbreviations (i.e. "DEX") should be explained in the text.
Lines 72-79, transfection chapter: Authors should describe the transfection method in details, that is a concentration of siRNA, dilution of lipofectamine and description of the medium used for actual transfection event. Supplementary data could be used for this purpose.
Similar attention is welcome for the next paragraph. Very informative would be to know the key features of those assays: the amount of RNA used for the transcription process, melting temperatures for PCR assays and others.
Line 93- WB paragraph: There is missing more detailed information about Ab's dilution factor, reaction conditions, i.e. temperature for overnight incubation.
Line 129., figure 1B: What is a possible explanation of the peak of mRNA level at 30th day? Is it correct to assume that 30th day is the highest, having presented curve fitting and time-point coverage?
Line 130, figure 2: What is the rationale behind a fact, that authors did not follow WB analysis of other presented genes? Correlation between mRNA and protein levels is a strong point of this article and make results very interesting and informative.
Line 158, figure 3:
- description of the experiments is slightly confusing, making it difficult to get the impression about the statistical power of each experiment.
- Did the authors check the uniformity of B-actin expression?
- pictures presented in 3e are lacking the description of the column, picture 3f is confusing with no description of which calculation method was preferred by the authors
- in 3g presented pictures are of poor quality in the term of size and resolution, which make impossible to check proper deposit distribution
Line 173: picture 4, especially 4c, 4d, 4e, shares the same remarks with picture 3
Line 181, discussion: In this paragraph, it would be more to add a few sentences more about possible mechanisms linking presented changes with AQP3 gene/protein expression, i.e. what would be a possible cause of a drop in mRNA expression of PPARgamma and other presented genes in AQP3-silencing experiments?
Line 224, figure 5: If so, is there any evidence of AQP3 acting as an active switch between those two states?
Author Response
April 4th, 2020
MS Number: Genes-763471
Title: AQP3 facilitates proliferation and adipogenic differentiation of porcine intramuscular adipocytes
Dear Editor and Reviewer,
We are grateful to you for all the comments and valuable suggestions. We have followed carefully all the comments and made changes to the MS according to your suggestions.
Here are the details of the response to reviewers’ comments. All the answers are marked in red in this revision letter.
Best wishes,
Xiao Li

Reviewer 2 Report
General comments:
Title: AQP3 facilitates proliferation and adipogenic differentiation of porcine intramuscular adipocytes
Content: Wang and colleagues performed a knockdown experiment on aquaglyceroporin AQP3 gene by siRNA and evaluated the effects on the suppression of the expression of AQP3 in intramuscolar precursor adipocytes (PIPAs) in pigs. Authors found a role of AQP3 in adipogenic differentiation and triglycerides content in intramuscular adipocytes
Finding new markers associated with the intramuscolar fat and marbling is an issue of interest for the pig production chain.
Major comments:
The introduction paragraph needs a global rearrangement. It lacks some essential topics:
Authors didn’t talk about any biochemical pathway, could authors describe the interactions among AQP3 and other aquaporins?
Could the Authors describe interactions among AQP3 and other adipogenic markers?
Regarding the material and methods paragraph, the references from line 84 to line 87 are not listed in the References at the end of the draft.
The Results paragraph lacks of some references (i.e. line 168 is stated that “accumulated studies shown…..cell-cycle” needs some reference).
Discussion
Line 194-195: The sentence with the ref 14 should be moved to the introduction.
Line 205-207: This sentence should be moved to the introduction.
Line 211-217: This sentence should be rearranged keeping all essential things related to the Authors’ results and moving the more introductive parts to the Introduction paragraph (i.e. “There are CpG sites…”).
Conclusions: In order to highlight the results I suggest to improve the conclusions and avoid the inclusion of the Fig.5 because it is not relevant, the concept should be explained in the text of the conclusions.
Author Response

(The authors gave the same response as above.)

Reviewer 3 Report
Wang and colleagues are proposing a muniscript entitled “AQP3 facilitates proliferation and adipogenic differentiation of porcine intramuscular adipocytes”. The draft aims to validate the involvement of the aquaglyceroporin 3 (AQP3) gene in the proliferation and in the triglygerides content of precursors of intramuscolar adipocytes collected from the longissimus dorsi muscle in pigs. The Authors performed a knockdown esperiment using siRNA technique and they obtained interesting results which seem to demonstrate the active role of AQP3 gene in the proliferation and differentiation of these adipocytes.
In my opinion, this manuscript needs to be improved but it deserves to be published after major revisions. I have some major comments, which require to be clarify, and some minor comments and suggestions.
My major contraints regard the organization of the manuscript. Even if I think that all methodologies are quite well explained and described, there are some essential missing information and topics in particular in the Introduction and in the Discussion paragraphs that need to be provided.
Introduction:
- It is not clear which are the molecular and biochemical links between the expression of AQP3 gene and the metabolic pathways related to fat deposition and triglygerides production. It is only mentioned that the AQPs gene family are water/glycerol channels, but it is not explained at all the pathways involved in this important phenotype.
- It is also not clear which are the interaction between AQP3 and other factors involved in the differentiation of adipocytes. These factors are then mentioned in the manuscript but they are never described in the Introduction, which shoud be the introductive part for a comprehension of a paper.
- In lines 47-48 it is mentioned the potential role of AQP3 in cancer related to cell proliferation. I think it is necessary to describe how AQP3 could be involved in cell proliferation, how it can act in the cell cycle (also because in the Materials and Methods and Results sections findings on the Cyclins molecules are described). Moreover, it should be also more comprehensive to cite other studies on the AQP3 role on cancer or metastasis in the Introduction paragraph and not in the Discussion one (see below).
- It is not clear why Authors decided to not compare their findings on intramuscolar adipocytes with other adipocytes in other tissues. I think that some comparisons should better design the paper and should give more informativeness to results.
Results, Discussion and Conclusions:
- Lines 168-169: this sentence of Results should be moved to the Introduction section and improved with other examples. Moreover references are missing.
- Lines 182-187: this part of the Discussion is a repetition. I suggest to reduce and summarize.
- Lines 188-196: this part of the Discussion should be moved in the Introduction section and these sentences could better explain the aim of the study.
- Lines 205-207: Again, this part is an introductive part and should be in the Introduction.
- Lines 212-217: Again, this is not a discussion of results, because this paper is not focused on cancer. Anyway, this part should be in the Introduction. Moreover, line 215-216: you mention the CpG sites in human AQP3 locus, what about pig? Are they comparable? Is this information essential to describe the role of AQP3?
- What about other tissues or organs with adipocytes? What Authors can retrieve about their findings? I think that some molecular and biochemical hypothesis based on the results are missing.
- Finally, the conclusions are a bit weak because Authors used word as “bland new target to adjust…”. It seems that the results are not “strong” enough. I would change this sentence.
The minor comments are:
- Some references are missing (lines 84-88) and they are mentioned in a different way comparing to other parts of the text.
- Line 44: the reference about “our previous RNA-seq screening” is missing.
- Figure 1 line 130: why in Material and Methods results there is no the description of RNA extraction in these other tissues (muscle, heart, liver, speen, lung, kidney)? From where these results come from?
- LIne 168: please substitute “is” with “are”.
- Line 200: please substitute “lead” with “leads”.
Author Response

(The authors gave the same response as above.)

Round 2
Reviewer 1 Report
Dear Authors,
Thank you for your reply, it cleared my view and answered all my substantive question.
I still have small technical "issues" I would like to have discussed by the Authors.
line 75: Please check if mesh description is in the right order (212 and 75).
line 119: "4oC", probably pdf conversion problem,
line 140: issue I didn't notice before, please refer to ImageJ software with a publication the provide instead of webpage ling
line 173: "FASN4d"
line 177: missing space before (Fig3 F)
line 180, figure 3g: In my opinion Oil red O pictures did not improve much in terms of resolution and picture detail quality. There is also scale bar added, which are not clear at all. I fully understand a fact, that this staining is only supportive of other presented experiments, however, even for a simple "yes/no" picture, this publication can benefit from the better quality picture.
Kind Regards,
Author Response
Dear Reviewer,
Thank you very much for the approval of our reply. Based on the round 2 of comments, we will continue to correct the manuscript as requested.
Here are the details of the response to reviewers’ comments. All the answers are marked in red in this revision letter.

Reviewer 2 Report
The Authors followed the suggestions and added new information in the text as requested.
They can maybe add the hypothesis that was conceptualized in the deleted Figure 5 in a sentence of the Conclusions.
Author Response
Dear Reviewer
Thank you for your suggestion. We supplement the conclusion.
Reviewer 3 Report
Wang and colleagues improved their manuscript, especially the Introduction part, as suggested. Moreover, the Authors provided better explanations of their results in the Discussion paragraph and followed the suggestions.
Author Response
Dear Reviewer
Thank you very much for your suggestions and confirmation of the manuscript.